# Towards Synergistic, Generalized and Efficient Dual-System for Robotic Manipulation

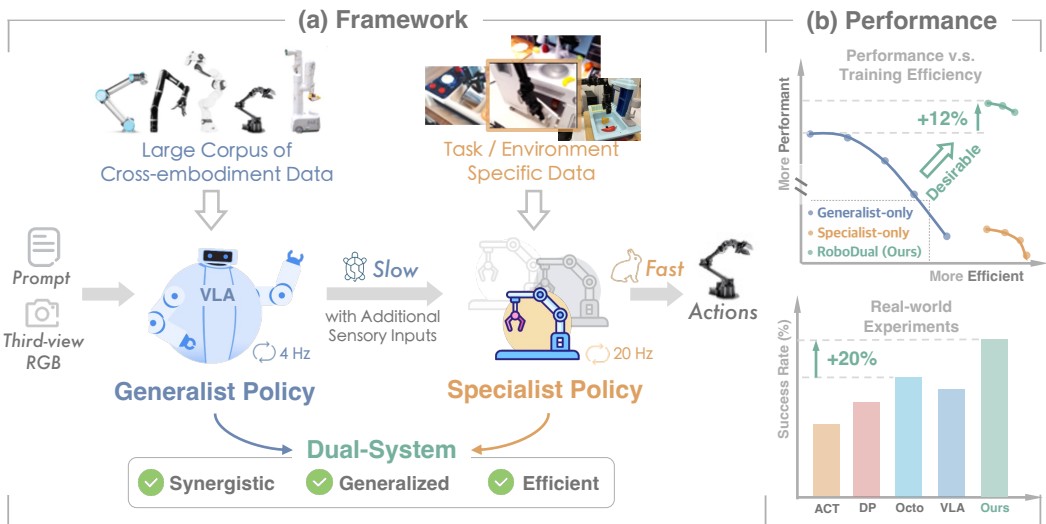

Figure 1: **Overview of `RoboDual`.** Our objective is to develop a synergistic dual-system framework which supplements the generalizability of large-scale pre-trained generalist with the efficient and task-specific adaptability of specialist. **(a)** The fast specialist policy achieves real-time and accurate control by aid of the slow yet generalized output from the generalist one trained with large-scale data. **(b)** `RoboDual` exhibits significant improvement in terms of performance and efficiency over a single standalone option and surpasses previous state-of-the-arts in the real-robot setting.

## ABSTRACT

The increasing demand for versatile robotic systems to operate in diverse and dynamic environments has emphasized the importance of a generalist policy, which leverages a large cross-embodiment data corpus to facilitate broad adaptability and high-level reasoning. However, the generalist would struggle with inefficient inference and cost-expensive training. The specialist policy, instead, is curated for specific domain data and excels at task-level precision with efficiency. Yet, it lacks the generalization capacity for a wide range of applications. Inspired by these observations, we introduce RoboDual, a synergistic dual-system that supplements the merits of both generalist and specialist policy. A diffusion transformer-based specialist is devised for multi-step action rollouts, exquisitely conditioned on the high-level task understanding and discretized action output of a vision-language-action (VLA) based generalist. Compared to OpenVLA, RoboDual achieves 26.7% improvement in real-world setting and 12% gain on CALVIN by introducing a specialist policy with merely 20M trainable parameters. It maintains strong performance with 5% of demonstration data only, and enables a 3.8× higher control frequency in real-world deployment. Code would be made publicly available. An anonymous real-robot demo is hosted at `https://robodual.github.io`.

## 1 INTRODUCTION

The pursuit of versatile and adaptive robotic intelligence has been a central objective in the robotics community for decades (Franklin, 1997; Kunze et al., 2018; Duan et al., 2022). Conventional

robot learning methods typically develop policies through datasets curated for the designated robot and its specific task. The resulting policy could be deemed as a *specialist*, including the popular ACT (Zhao et al., 2023) and Diffusion Policy (Chi et al., 2023). It exhibits high precision in dedicated scenarios and tasks, and yet often demonstrates limited generalization ability (Brohan et al., 2023a). As robots are increasingly employed in open-ended and multi-task environments, the demand for systems capable of handling diverse tasks and adapting seamlessly across various embodiments has surged. This has fueled the development of the *generalist*, such as RT-2 (Brohan et al., 2023a) and Octo (Ghosh et al., 2024). They leverage extensive, heterogeneous datasets to enhance cross-domain generalizability and aim to transfer web knowledge to robotic control. Recent advances on Vision-Language-Action (VLA) approaches (Li et al., 2024; Brohan et al., 2023a; Kim et al., 2024; Covariant, 2024) exemplify the potential of generalist policy to meet the ever-evolving demands. VLAs integrate vast cross-embodiment data with pre-trained large (vision-)language models, facilitating capabilities such as common-sense reasoning and instruction following (Zhao et al., 2024).

While VLA-based generalists excel at knowledge transfer and generalization across diverse scenarios, several limitations remain: **1)** They cannot be directly deployed to new embodiments or environments out-of-the-box without adaptation (Wang et al., 2024b). The finetuning process is more data and training intensive compared to specialist policies (Fu et al., 2024). **2)** Though VLAs are skilled in high-level decision making, their large model nature leads to extremely high inference latency (Brohan et al., 2023a). This pivotal bottleneck makes them unsuitable for fine-grained control in dynamic environments. **3)** Current generalist models support single-frame RGB observations only, which, while enabling training on larger-scale datasets, restricts their effectiveness in tasks where additional sensory inputs such as depth or tactile feedback play pivotal roles. In the meantime, incorporating these extra modalities requires resource-intensive re-training (Han et al., 2024) and runs the risk of catastrophic forgetting (Zhai et al., 2023b).

Significant efforts, such as model quantization and fine-tuning of the generalist with multimodal data, have been made to address the aforementioned limitations (Kim et al., 2024; Zhen et al., 2024). Nonetheless, these approaches still face inevitable performance declines or data scarcity-related issues. This raises the question of whether it is sufficient to rely solely on enhancements to generalists to resolve these challenges. We recall that a generalist model offers broad generalizability and benefits from web-scale pre-training, while a specialist policy is competent at efficiency and fast adaptation to specific tasks. Based on the insights, we propose a generalized and efficient framework in which both policies complement each other for improved manipulation, as illustrated in Figure 1(a). Our work introduces a novel dual-system[1] synergy approach, namely **RoboDual**. It is designed to harness the advantages of both parties and facilitate the practical deployment of large generalist policies.

We start with the large-scale pre-trained OpenVLA (Kim et al., 2024) to establish our generalist policy. For seamless cooperation between the two models, we implement the specialist model as a lightweight and scalable diffusion transformer (Peebles & Xie, 2023) policy. The specialist learns the multimodal action distribution by utilizing any sensory inputs and the generalist outputs adaptively through a unified conditioning mechanism. Latent representations and discretized action outputs from the generalist enable our specialist to adapt to new tasks or environments efficiently with minimal data and training costs. During inference, the generalist provides deliberate yet comparatively slower conditioning, which supports multistep roll-outs of the fast-reacting specialist to achieve precise and generalized control. In this way, RoboDual is rendered with high-level task understanding and generalizability from the generalist, combined with efficient action refinement of the specialist, achieving outstanding performance across a diverse array of tasks. As demonstrated in Figure 1(b), it realizes a 12% performance gain over the generalist-only variant on CALVIN (Mees et al., 2022b) with minimal training cost. In real-robot setting, RoboDual outperforms both specialist and generalist baselines by a significant margin. To summarize, our contributions are threefold:

- We introduce a novel approach that integrates generalist and specialist policies into a synergistic framework, dubbed as RoboDual, following a dual-system spirit. Our methodology leverages the merits of each party and paves the way for the practical application of VLAs to dexterous tasks.

---

[1]In cognitive science, the dual-system framework distinguishes between System-1 that engages in rapid, automatic processing, and System-2, which involves slow, deliberate thinking with intentional effort (Kahneman, 2011). By analogy, it can be suggested that our specialist model may operate in a manner akin to System-1, while the generalist model might reflect the characteristics of System-2.

- We propose a diffusion transformer-based specialist policy that enables real-time control, adaptively conditioned by generalist outputs and various sensory inputs. The framework facilitates the flexible integration of diverse modalities and allows for the deconstruction of the two models on the aspect of training data, thereby enhancing their individual strengths and capabilities.
- We demonstrate that the dual-system approach surpasses both specialist- and generalist-only models in various tasks, through extensive real-world and simulation experiments.

## 2 RELATED WORK

**Generalist policy.** The RT-X series of works (Brohan et al., 2023b;a) have sparked significant progress in the development of multi-task generalist policies (Yang et al., 2023; Ghosh et al., 2024) by leveraging extensive cross-embodiment datasets (Padalkar et al., 2024; Walke et al., 2023; Khazatsky et al., 2024). Octo (Ghosh et al., 2024) employs a transformer-based policy trained on 800k trajectories from the Open-X-Embodiment dataset (Padalkar et al., 2024), enabling flexible fine-tuning for novel robotic configurations. OpenVLA (Kim et al., 2024), which represents the most advanced generalist policy to date, directly integrates pre-trained vision-language models to generate robotic actions by treating them as tokens within the language model's vocabulary. Unlike Octo, which relies solely on a pre-trained language encoder T5 (Raffel et al., 2020) and derives generalization primarily from large-scale in-domain policy training, OpenVLA harnesses world knowledge from a much broader vision-language dataset. Despite its demonstrated generalizability, OpenVLA's model size, which comprises billions of parameters, hinders both data and inference efficiency. Hence it would confine the deployment on heterogeneous robot setup as a generalist policy. In contrast, our approach introduces a novel and cost-effective dual-system framework to address these caveats, instead of developing another larger generalist model reliant on extensive datasets.

**Specialist policy.** We regard specialist models as policies specifically trained to execute a narrow set of tasks or functions with high precision (Goyal et al., 2023; 2024). These models typically utilize curated datasets tailored to their specific applications (Zhao et al., 2023; Chi et al., 2023). While many specialist policies excel in few-shot imitation learning, they often lack the integration of language inputs, necessitating the training of distinct models for different tasks. A notable trend in prior studies is the reliance on 3D representations (Shridhar et al., 2023; Gervet et al., 2023; Ze et al., 2023; Yan et al., 2024; Goyal et al., 2024) to improve performance on low-dimensional control tasks. Transformer-based models have emerged as powerful tools for extracting multimodal features, enhancing manipulation capabilities through their flexibility in processing heterogeneous observations (Kim et al., 2021; Dasari & Gupta, 2021; Goyal et al., 2023; Simeonov et al., 2023). The recent development of the Diffusion Policy (Chi et al., 2023), along with subsequent works (Ze et al., 2024; Prasad et al., 2024), has proven effective in managing multimodal action distributions for robotic manipulation while exhibiting improved training stability. In our work, the specialist model is designed as a scalable diffusion transformer (Peebles & Xie, 2023) policy, adept at handling multimodal inputs and generalist outputs as conditioning in a unified framework. Furthermore, we demonstrate how collaboration with a generalist model enhances generalization and enables the execution of multi-instruction tasks that previous specialist-only models struggle to address.

**Hierarchical control with LLMs.** The rise of Large Language Models (LLMs) and their ability to interpret prompts and perform reasoning has sparked interest in their application to robotics (Wang et al., 2024a). A key area is high-level task planning, where LLMs decompose tasks using natural language, as demonstrated in SayCan (Ahn et al., 2022), PaLM-E (Driess et al., 2023), and HiP (Ajay et al., 2024). This enables robots to translate abstract commands into concrete actions. This is followed by works using structured code for control (Liang et al., 2023; Singh et al., 2023), which map user instructions into executable programs. Another line of work, including RoboFlamingo (Li et al., 2024) and LCB (Shentu et al., 2024), involves hierarchical control via latent space representations, where they employ an additional action decoder network on top of LLM latent outputs to regress actions directly. Recent works also explore action tokenization through VQVAE techniques (Van Den Oord et al., 2017) to better bridge LLMs and actions (Wang et al., 2024c; Szot et al., 2024). Although our framework shares a similar hierarchical philosophy, it does not explicitly decompose tasks and does not require end-to-end optimization of decoupled policies. Instead, we utilize both discretized outputs and latent embeddings as "interfaces" to connect two models, rather than pre-defined low-level skills. The generalist and specialist can be decoupled on the aspect of training data to develop their respective capabilities, which also brings flexibility to utilizing various modal inputs.

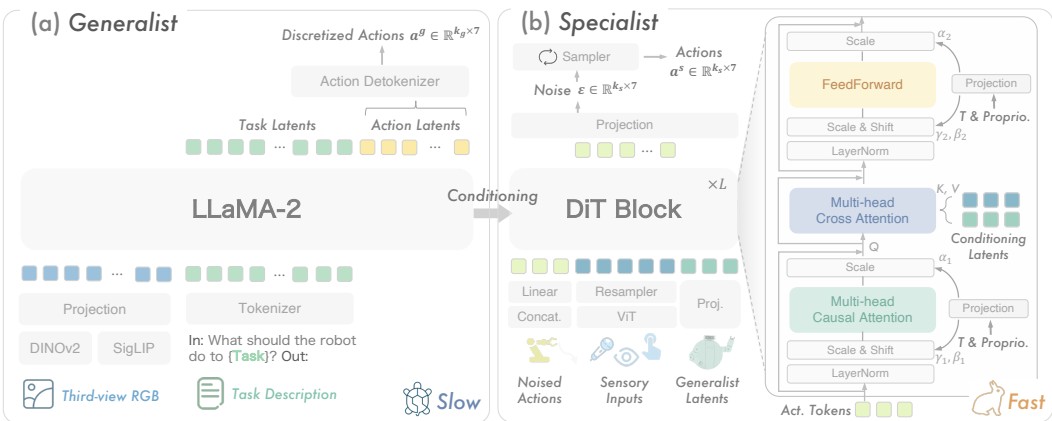

Figure 2: **The overall architecture of `RoboDual`. (a) Generalist**. The generalist takes as inputs RGB images and language prompts, generating conditioning sources for the specialist model, including latent representations and discretized actions. **(b) Specialist**. Comprising stacked Diffusion Transformer (DiT) blocks, the specialist is conditioned on multiple sensory inputs and the generalist's output through a cross-attention mechanism. It predicts noise injected into ground truth actions, providing fast, precise control by leveraging the slower, high-level guidance of the generalist.

## 3 ROBODUAL: GENERALIST-SPECIALIST SYNERGY

Our goal is to develop a synergistic and generalized framework that leverages the strengths of both generalist and specialist policies while simultaneously addressing their respective limitations beyond mere integration. We introduce the generalist policy, which forms a meticulous and robust planner in our framework in Section 3.1. In Section 3.2, we provide design principles of the specialist policy, which is optimized for precise, real-time control and allows unified conditioning. Finally, we describe our training and inference protocols in Section 3.3. The systematic diagram is illustrated in Figure 2.

### 3.1 GENERALIST: AN AUTO-REGRESSIVE VISION-LANGUAGE-ACTION MODEL

Figure 2(a) demonstrates our generalist's architecture. Our generalist model is built upon Open-VLA (Kim et al., 2024), a 7B-parameter autoregressive vision-language-action model trained with a large corpus of robotic manipulation data, including Open-X-Embodiment (Padalkar et al., 2024), Bridge V2 (Walke et al., 2023), DROID (Khazatsky et al., 2024), *etc*. The generalist model follows the architecture of Prismatic-7B (Karamcheti et al., 2024) vision-language model (VLM), which consists of a fused visual encoder from different backbones (Zhai et al., 2023a; Oquab et al., 2024), a projection layer to align visual embeddings with language modality, and a large language model LLaMA2 (Touvron et al., 2023). Despite extensive training on a large-scale cross-embodiment dataset, OpenVLA is not capable of functioning in a zero-shot manner in new environments or embodiments (Wang et al., 2024b). Adaptation to our specific robotic setup and test environments (with novel coordination system, camera angle, *etc*.) remains necessary, which we accomplish through LoRA (Hu et al., 2022) fine-tuning. Nonetheless, we intend to leverage the massive pre-trained knowledge embedded in OpenVLA to endow our dual-system framework with certain generalizability.

**Autoregressive generation of action chunking.** Following RT-2 (Brohan et al., 2023a) and Open-VLA (Kim et al., 2024), we map the least used 256 words in the LLaMA tokenizer vocabulary into uniformly distributed action bins within $[-1, 1]$. This approach allows us to detokenize language tokens into discretized actions based on their corresponding indices in the vocabulary. The generalist model decodes every degree-of-freedom of actions in an auto-regressive manner, where the decoding for the current token is dependent upon input prompts and previously decoded tokens. We further extend the original OpenVLA to predict action chunks with a temporal length of $k_g$. This longer-range planning on the generalist side enhances its own ability to capture non-Markovian behavior in human demonstrations, and also facilitates more informative conditioning provided to the specialist model. The action output corresponding to each time step is separated by `[space]` token in the tokenizer vocabulary. However, action chunking increases the inference latency of VLA due to the generation

of a greater number of tokens. This further prompts the need for a specialist model that runs at a higher frequency between consecutive VLA outputs to achieve more responsive control.

## 3.2 SPECIALIST: A CONTROLLABLE DIFFUSION TRANSFORMER POLICY

Built on top of a pre-trained generalist policy, the specialist is to achieve improved performance while reducing control latency, even with limited training data and compute. To fully exploit the multimodal sensory inputs necessary for effective manipulation, as well as the privileged knowledge from the generalist policy, we design the specialist based on Diffusion Transformer (DiT) (Peebles & Xie, 2023), to perform controllable action sequences denoising.

**Base architecture.** Figure 2(b) illustrates the architecture of our specialist model, which is primarily composed of stacked DiT blocks. Each block includes a causal self-attention layer to process temporal actions, a cross-attention layer to fuse information, and a point-wise feedforward network that performs non-linear transformations. Drawing parallels with image diffusion models (Saharia et al., 2022), we treat a 7-DoF action as a pixel with seven channels, which is linearly projected into a single token and processed by the diffusion model. This formulation facilitates a seamless temporal expansion of action tokens, enabling action chunk prediction (Zhao et al., 2023) with a flexible temporal length of $k_s$. We employ Vision Transformers (ViT) (Dosovitskiy et al., 2021) as generalized sensory encoders to encode all possible input modalities (*e.g.*, RGB, depth, and tactile), with minor modifications on the patchify layer given different number of channels. A DINO (Caron et al., 2021) pre-trained model is leveraged to encode the RGB inputs, which is frozen during training. Encoders for other modalities are constrained to six layers with a hidden size of 256 to ensure efficiency. Beyond what we have explored, our framework is also adaptable to non-image inputs that can be encoded into a sequence of embeddings.

**Action denoising with multimodal conditioning.** The specialist model leverages multiple sources of conditioning and their corresponding conditioning approaches to enhance decision-making: **1)** proprioceptive states (Proprio.) of the robot, **2)** multimodal sensory inputs, **3)** generalists' discretized action outputs, and **4)** latent representations (refer to Figure 2(a)) from the generalist model. Each source contributes distinct information, facilitating a more informed and robust policy.

The *proprioceptive states* are processed through a two-layer MLP and combined with a time-step embedding to enable adaptive sample-wise conditioning. Beyond regressing $\gamma$ and $\beta$ parameters for adaptive layer normalization (Perez et al., 2018), a scaling parameter $\alpha$, is introduced in the residual connections to ensure stable conditioning and improve training robustness (Peebles & Xie, 2023).

For *sensory inputs*, we incorporate a perceiver resampler (Alayrac et al., 2022), consisting of a multi-head attention pooling module followed by an MLP layer, to selectively distill key features from observation embeddings generated by ViTs while reducing token length. Specifically, we employ eight learnable queries for every sensory input. The resampler preserves performance and accelerates the multistep denoising process, particularly when dealing with multi-source inputs advantageous for manipulation tasks, such as multiview observations, historical frames, and multimodal data.

To condition the specialist on the *discretized actions* from the generalist, we concatenate them with the noised action of the corresponding time step, and project the concatenated inputs into a shared latent space through linear layers. This approach is inspired by video prediction models (Blattmann et al., 2023), which concatenate initially known frames with noised inputs to predict future states.

Conditioning our specialist model on the *task and action latents* derived from the generalist involves utilizing linear projection on the generalist tokens to align their hidden spaces. Despite simplicity, it is parameter-efficient and preserves the original positional encoding within VLA. Finally, the projected generalist latents, along with the resampled observation embeddings, are concatenated and utilized as keys and values in the cross-attention layer. Our diverse conditioning enables the specialist model to process comprehensive contextual data effectively, prompting more informed decision-making.

Given that the generalist and specialist models operate asynchronously during inference (with a single generalist inference supporting multiple specialist rollouts), we implement a shifted-window conditioning mechanism. Specifically, following $\tau_s$ steps of specialist inference, only the most recent $k_g - \tau_s$ generalist actions are sampled as conditioning before a second update. We also apply this mechanism as an augmentation during training to ensure latency robustness of RoboDual. The

specialist model is optimized to denoise future trajectories by conditioning on the lagged generalist's output, with the specialist's observation input leading the generalist's by $\tau \in [0, k_g]$ steps.

### 3.3 TRAINING & INFERENCE PROTOCOL

**Generalist training.** Unlike recent studies (Li et al., 2024; Szot et al., 2024) that directly applies action regression loss to the output tokens of VLMs, we follow OpenVLA (Kim et al., 2024) and use discrete token prediction, which naturally aligns the next-token prediction approach of decoder-only LLMs (Chen et al., 2021). The model $g_\phi$ is fed with prompts $\mathbf{p}$ and prefixes of the ground truth actions $a_{<i}$, and trained to minimize the sum of next-token negative log-probabilities:

$$\mathcal{L}_{\text{gen}} = \mathbb{E}_{\mathbf{p}, a_{<i}} \left[ -\sum_{i=1}^{N_a} \log g_\phi(\hat{a}_i = a_i \mid \mathbf{p}, a_{<i}) \right], \tag{1}$$

where $N_a$ represents the total length of action tokens. During the inference stage, the generalist decodes $\hat{a}_i$ based on previously decoded tokens $\hat{a}_{<i}$, instead of ground truth actions.

**Specialist training.** Following Diffusion Policy (Chi et al., 2023), we train our specialist with an action denoising objective. Given an action trajectory of temporal length $k_s$ from dataset $a_0 \sim \mathcal{D}^a$, randomly sampled noise $\epsilon \sim \mathcal{N}(0, \mathbf{I})$, and an arbitrary timestamp $t \sim \mathcal{U}(1, T)$, where $t \in \mathbb{Z}, T = 100$, the forward diffusion process is formulated in closed form as $a_t = \sqrt{\overline{\alpha}_t} a_0 + \sqrt{1 - \overline{\alpha}_t}\epsilon$. $\overline{\alpha}_t$ denotes noise schedule that performs one-step noise adding (Ho et al., 2020). We optimize the following training objective to train the specialist model $\pi_\theta$ as follows:

$$\mathcal{L}_{\text{spec}} = \mathbb{E}_{t, c, a_0, \epsilon} \left[ \|\epsilon - \pi_\theta(\sqrt{\overline{\alpha}_t} a_0 + \sqrt{1 - \overline{\alpha}_t}\epsilon, \ c, \ t)\|^2 \right], \tag{2}$$

where $c$ denotes the set of conditioning sources. We explore training a lightweight specialist model from scratch, conditioned on a pre-trained generalist, and promote synergistic interactions between the two systems. Introducing merely 20M trainable parameters and 8 GPU-hours of training with our specialist model, the resulting dual-system demonstrates a more significant performance improvement compared to the gains achieved from several days of additional training on the VLA alone (17% *v.s.* 10%). We delve into this in Section 4.4. Further details regarding training hyperparameters and architecture design are provided in Appendix B.

## 4 EXPERIMENTS

We conduct extensive experiments to evaluate the performance of our method and to highlight its notable attributes concerning generalizability, efficiency, and adaptability. We intend to study the following research questions: **I.** (Section 4.2) Could RoboDual demonstrate higher success rates in simulation and real-world tests compared to previous methods? **II.** (Section 4.3) Does RoboDual perform generalizable manipulation? **III.** (Section 4.4) How is the adaptation and inference efficiency of RoboDual? **IV.** (Section 4.5) What are the key factors contributing to the dual-system synergy?

### 4.1 EVALUATION SUITE

**Simulation experiments on CALVIN.** CALVIN (Mees et al., 2022b) is a widely recognized simulation benchmark for assessing long-horizon language-conditioned manipulation tasks. Our objective is to demonstrate the generalizability of our system in multitask learning using free-form language instructions. Additionally, we investigate how the specialist model can leverage multiple input modalities, beyond the third-view RGB input of the generalist, to enhance manipulation performance. Further information regarding the benchmark and implementation details are provided in Appendix A.

**Real-world robot experiments.** All real-world experiments are conducted with an ALOHA platform featuring a 7-DoF action space and a third-view RGB camera. We evaluate policies on both single-instruction tasks ("Lift the pod did", "Pour shrimp into bowl", and "Push block Left") and multi-instruction tasks ("Put <obj> into basket" and "Knock <obj> over"). Additionally, we propose a comprehensive set of evaluation tasks that cover various axes of generalization: 1) position variation, 2) visual distractions, 3) unseen background, and 4) novel objects. Each task is collected with teleportation for 20-120 demonstrations based on their complexity. To establish our baselines, we take the most advanced and widely adopted specialist policies, ACT (Zhao et al., 2023) and

Table 1: **Language-conditioned visuomotor control on CALVIN ABC→D.** We report success rates along with the average length of completed tasks (out of the whole 5 tasks) per evaluation sequence. *Lang* and *All* denote whether models are trained only with the subset vision-language data pairs.

| Method | Train sets | Task completed in a row (%) ↑ | | | | | Avg. Len. ↑ |
|---|---|---|---|---|---|---|---|
| | | 1 | 2 | 3 | 4 | 5 | |
| MCIL (Lynch & Sermanet, 2021) | All | 30.4 | 1.3 | 0.2 | 0.0 | 0.0 | 0.31 |
| HULC (Mees et al., 2022a) | All | 41.8 | 16.5 | 5.7 | 1.9 | 1.1 | 0.67 |
| RT-1 (Brohan et al., 2023b) | Lang | 53.3 | 22.2 | 9.4 | 3.8 | 1.3 | 0.90 |
| MDT (Reuss et al., 2024) | Lang | 61.7 | 41.6 | 23.8 | 14.7 | 8.7 | 1.54 |
| RoboFlamingo (Li et al., 2024) | Lang | 82.4 | 61.9 | 46.6 | 33.1 | 23.5 | 2.48 |
| SuSIE (Black et al., 2024) | All | 87.0 | 69.0 | 49.0 | 38.0 | 26.0 | 2.69 |
| GR-1 (Wu et al., 2024) | Lang | 85.4 | 71.2 | 59.6 | 49.7 | 40.1 | 3.06 |
| 3D Diffuser Actor (Ke et al., 2024) | Lang | 92.2 | 78.7 | 63.9 | 51.2 | 41.2 | 3.27 |
| RoboDual (Ours) | Lang | **94.4** | **82.7** | **72.1** | **62.4** | **54.4** | **3.66** |

Diffusion Policy (Chi et al., 2023), alongside generalist models, Octo (Ghosh et al., 2024) and OpenVLA (Kim et al., 2024), for comparative analysis. Specialists are trained in a single-task manner, while the generalists are first trained with the combination of all tasks and then tuned on specific scenarios to optimize their performance. For a clearer comparison, we implement our method by first training the generalist across all tasks, followed by training the specialist separately using either task-specific data (Ours-single-task) or multi-task data (Ours-multi-task). To enable specialist baselines, which do not use language inputs, to effectively learn multi-instruction tasks, we incorporate FiLM conditioning (Perez et al., 2018) into the visual backbone as RT-1 (Brohan et al., 2023b). For all tasks, we report the average success rate over 15 independent runs. More details are in Appendix A.

## 4.2 COMPARISON TO STATE-OF-THE-ARTS

**CALVIN benchmark.** We compare the performance of RoboDual with other state-of-the-art methods on CALVIN ABC→D. The results are given in Table 1. We yield an improvement from **3.27** to **3.66** on the average length of completed tasks. The success rate of accomplishing consecutive 5 tasks is elevated by **13.2%**. Additionally, we further investigate the robustness to free-form task instructions of various methods, as shown in Table 2. We incorporate

Table 2: **Evaluations on robustness to free-form language instructions.** RoboDual shows exceptional instruction-following capability.

| Method | Task completed in a row (%) ↑ | | | | | Avg. Len. ↑ |
|---|---|---|---|---|---|---|
| | 1 | 2 | 3 | 4 | 5 | |
| RoboFlamingo | 63.0 | 33.0 | 16.4 | 8.6 | 3.6 | 0.40 |
| 3D Diffuser Actor | 65.2 | 39.1 | 20.3 | 11.7 | 6.1 | 1.42 |
| LCB | 73.6 | 50.2 | 28.5 | 16.0 | 9.9 | 1.78 |
| RoboDual (Ours) | **91.8** | **81.8** | **68.5** | **57.8** | **48.8** | **3.47** |

RoboFlamingo and LCB (Shentu et al., 2024), both of which also utilize LLMs (MPT-1B (MosaicML, 2023) and LLaVA-7B (Liu et al., 2024) respectively), as our baseline approaches. All methods are trained exclusively on the ABC split using the original language annotations and are evaluated with GPT-4 enriched ones. While the performance of baseline methods decreases significantly compared to their results in Table 1, our method exhibits minimal impact and nearly doubles the average length compared to LCB. This improvement can be attributed to both the semantic understanding capability of the generalist and the specialist model's robustness to variations in conditioning latents.

**Real-world experiments.** The results are presented in Figure 3. The state-of-the-art specialist policy, Diffusion Policy, achieves smoother control with a high success rate on more dexterous, yet narrowly defined tasks. However, it struggles significantly with multi-instruction tasks, achieving only a 20% success rate in "Put <obj> into basket." In contrast, OpenX-pretrained generalist models (Octo and OpenVLA) perform better on diverse tasks involving multiple objects and requiring language conditioning. Regarding OpenVLA, its data and inference efficiency could be primary bottlenecks that constrain its overall performance. In the "Push block left" task, OpenVLA overfits to specific trajectories, despite demonstrations involving block placements in three distinct positions. Its high inference latency also induces jittering and pauses, undermining performance in dexterous control tasks. RoboDual harnesses the high-level reasoning capabilities of the generalist to guide the specialist in executing smooth control, demonstrating strong performance across both single- and multi-instruction tasks. Overall, RoboDual exhibits an improvement of **+20%** compared to the most competitive baseline and consistently achieves a minimum success rate of **60%** across all tasks.

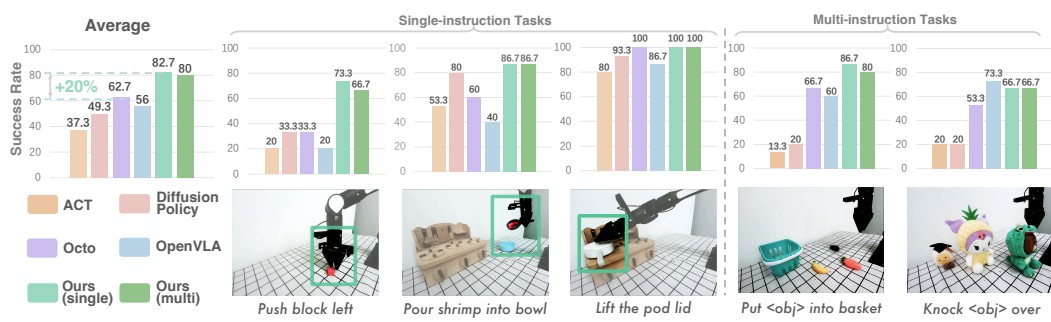

Figure 3: **Real-world robot experiments.** We report the success rates of each method on three single-instruction tasks and two multi-instruction tasks, along with the aggregate performance. RoboDual outperforms all specialist and generalist baselines by a notable margin across all tasks.

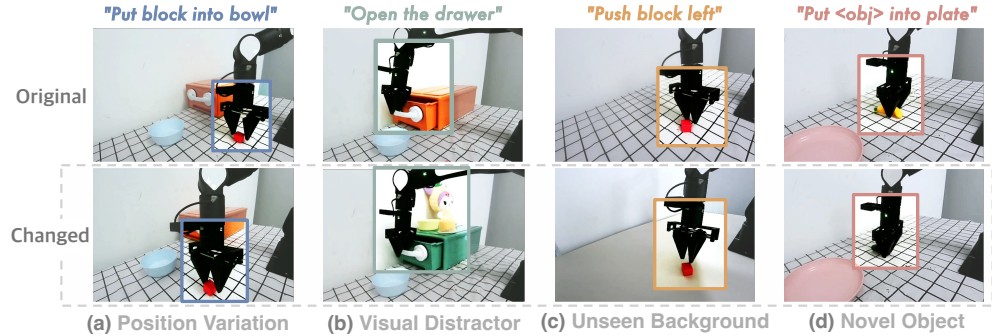

Figure 4: **Setting on generalizability evaluation.** We evaluate four axes of generalizability with different tasks: **(a)** Position Variation: The red block will be randomly placed in a $20cm \times 10cm$ area; **(b)** Visual Distractor: We change the color of drawer, with added plush toys, yellow bowls and clay as distractors; **(c)** Unseen Background: We replace the checkered tablecloth with a solid white one; and **(d)** Novel Object: Manipulated objects are replaced with unseen ones (banana $\rightarrow$ eggplant).

Table 3: **Generalizability evaluation.** RoboDual excels at all evaluated tasks that require generalization capability from high-level semantic understanding to low-level position variation.

| Framework | Method | Position Variation | Visual Distractor | Unseen Background | Novel Object | Average ↑ |
|---|---|---|---|---|---|---|
| Specialist | ACT (Zhao et al., 2023) | 46.7 | 26.7 | 0 | 13.3 | 21.7 |
| | Diffusion Policy (Chi et al., 2023) | 53.3 | 40.0 | 26.7 | 40.0 | 40.0 |
| Generalist | Octo (Ghosh et al., 2024) | 20.0 | 60.0 | 6.7 | 6.7 | 23.4 |
| | OpenVLA (Kim et al., 2024) | 26.7 | 73.3 | 20.0 | 46.7 | 41.7 |
| RoboDual | Ours-single-task | **93.3** | **80.0** | **60.0** | 46.7 | **70.0** |
| | Ours-multi-task | 86.7 | 73.3 | 53.3 | **60.0** | 68.3 |

## 4.3 GENERALIZABILITY EVALUATION

We investigate the generalizability of RoboDual and baseline methods from four different aspects, as illustrated in Figure 4. Detailed quantitative results are given in Table 3. In task "Put block into bowl" that requires position generalization, specialist policies and Octo exhibit a noticeable bias towards certain tested positions, likely due to the inherent imbalance in the training data. Although OpenVLA can move in the correct direction, its control frequency prevents swift adjustments to the end effector, often causing the object to be pushed away before it can be grasped. Conversely, both the multi-task and single-task learning variants of RoboDual demonstrate strong performance on this challenging task. It is also noted that RoboDual demonstrates error-correction capabilities by attempting to re-grasp the block following a failed initial attempt, as shown in our video demos. Moreover, thanks to the large, pre-trained VLMs leveraged, RoboDual outperforms both ACT and Diffusion Policy in tasks necessities high-level generalizability (*e.g.,* "visual distractor" and "novel object"). These results demonstrate RoboDual's superior ability to generalize across various scenarios.

## 4.4 EFFICIENCY ANALYSIS

**Training efficiency.** We first investigate the efficient adaptation of RoboDual to new environments, as illustrated in Figure 5. The "generalist-only" and "specialist-only" variants are implemented precisely as utilized within our framework to ensure a fair comparison. To facilitate multitask learning on CALVIN using our specialist model, we employ T5-xxl (Raffel et al., 2020) to encode task instructions and concatenate the language embeddings with visual observations for conditioning. The performance of the generalist-only variant stabilizes at 3.27 after approximately 1,400 GPU hours of training on CALVIN. Based on this, RoboDual, which further incorporates a specialist model with only 20M trainable parameters, enhances performance to 3.52 following just one hour of training on a node equipped with eight A100 GPUs. Our approach ultimately improves the performance of a fully-trained generalist by an additional **12%**. Notably, when applied to an inadequately trained generalist, just one hour of adapta-

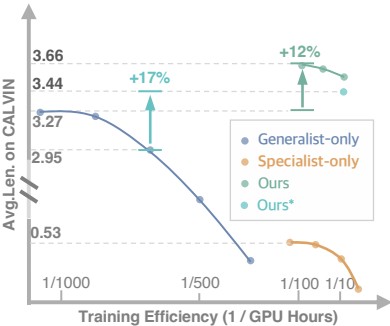

Figure 5: **Training efficiency.** We report the training duration (GPU hours) of the different methods and the model performance in different training phases. RoboDual achieves notable performance gains with minimum training time.

tion with RoboDual achieves greater improvement than several days of further training with the VLA alone (3.44 *v.s.* 3.27, denoted as **Ours**[*] in Figure 5). These results demonstrate that RoboDual is a cost-effective approach that provides substantial performance gains with minimal training costs.

**Data efficiency.** We then investigate how the specialist model can efficiently adapt to new environments and improve the overall performance with limited in-domain data. Results on CALVIN are listed in Table 4(a). We take RoboFlamingo, which also utilizes a large VLM, to conduct comparative analysis. We initialize RoboFlamingo with its official checkpoint trained on the full set of CALVIN dataset and reinitialize its LSTM-based action decoder head to train it with a sub-proportion of data.

Table 4: **Data efficiency.** We explore the adaptability of our specialist model to novel tasks with constrained data availability. RoboDual's performance advantage becomes more pronounced with limited data. *R.F.*: RoboFlamingo.

(a) **CALVIN benchmark.**

| Data | Avg. Len. | |
| Scale | R.F. | RoboDual |
| --- | --- | --- |
| 5% | 1.35 | **3.59** |
| 10% | 1.71 | **3.62** |
| Full | 2.48 | **3.66** |

(b) **Real-world experiments.**

| No. | Success Rate | | |
| Demos | ACT | D.P. | RoboDual |
| --- | --- | --- | --- |
| 5 | 0 | 20.0 | **73.3** |
| 10 | 6.7 | 20.0 | **80.0** |
| 100 | 46.7 | 53.3 | **93.3** |

While RoboFlamingo's results diminish by nearly half when utilizing only 5% of demonstrations, RoboDual exhibits robust performance with limited data and maintains an average length of 3.59, demonstrating exceptional data efficiency. The superiority of our method is further evidenced by real-world experiments. As presented in Table 4(b), we select "put block into bowl" as a exemplar task. Training with just five demonstrations, RoboDual achieves a notable success rate of **73.3%**, which is more than three times the performance of the diffusion policy (D.P.). In contrast, ACT completely fails to grasp the block with such limited demonstrations. Additionally, our method demonstrates the ability to succeed in novel block positions which are not included in the training set with only 5 episodes. Such an observation indicates that the specialist can effectively leverage privileged knowledge from the generalist and extrapolate with limited data.

**Inference latency analysis.** In real-world experiments, we constrain the generalist model to produce a single action (specifically the 8th action), which is then followed by eight steps of rapid specialist inference with a latency of 0.035 seconds. Correspondingly, RoboDual achieves a control frequency of **15 Hz** in our real-world setup using NVIDIA A5000 Ada GPUs, facilitating deployment in more dexterous tasks. Notably, inference latency is a primary factor contributing to the performance degradation of OpenVLA. Operating at only **3.9 Hz** within our system, it significantly alters the system dynamics compared to the 20 Hz non-blocking controller used in our real-world tasks. It can be observed that OpenVLA often struggles to adjust the end effector precisely before initiating a grasp, resulting in lower success rates for tasks requiring fine control, such as "Put block into bowl" (Table 3). Therefore, we believe that RoboDual is an effective and simple solution for deploying large VLA models in diverse robotics setups. We include video demos in our anonymous project page to offer a more intuitive glimpse into RoboDual's performance in real-world scenarios.

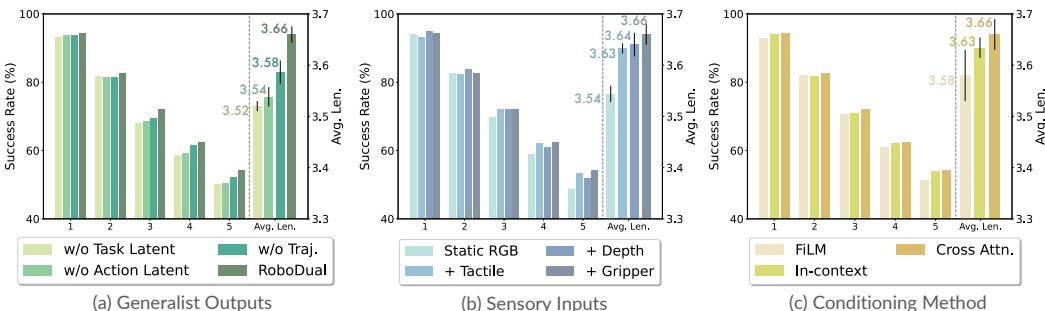

Figure 6: **Ablations on the factors affecting system synergy.** We investigate: **(a)** The importance of each output from the generalist as conditioning sources, **(b)** The effectiveness of incorporating additional sensory inputs, and **(c)** Comparison between different conditioning methods.

### 4.5 Ablation Study

The competitive results presented above position our approach favorably compared to specialist and generalist-only policies. In the following, we examine factors that pay credit towards a more desirable synergistic framework for RoboDual.

**Generalist outputs as conditioning.** Figure 6(a) describes that each conditioning source from the generalist model contributes to synergy and enhances overall performance. Although discretized trajectories may not be perfect, they offer plausible directions that can be iteratively refined by the specialist model, resulting in an improvement of 0.8. Furthermore, both the action and task latents (refer to Figure 2) encapsulate high-level task understanding, which is crucial for multi-task learning. The absence of task latents solely leads to a performance decline of 0.14 in average length.

**Additional sensory inputs.** Incorporating additional sensory inputs into the specialist model for specific scenarios proves to be a cost-effective strategy. Our approach eliminates the necessity for further fine-tuning of the VLA, without compromising the inference efficiency of the specialist model. As demonstrated in Figure 6(b), RoboDual leverages additional modalities (*e.g.,* depth and tactile) and extra viewpoints (*e.g.,* gripper camera) effectively to enhance overall performance.

**Conditioning method.** The firm bridges connecting generalist and specialist models are constructed through stable conditioning mechanisms. We assess three well-established methods: FiLM (Perez et al., 2018), in-context conditioning (Wang et al., 2023), and cross-attention. Results are shown in Figure 6(c). Cross-attention based method performs slightly better than in-context conditioning while also being more computationally efficient. Therefore, it is selected as our final approach.

## 5 Conclusion and Future Work

We present RoboDual, a synergistic dual-system for robotic manipulation that capitalizes the generalizability of Vision-Language-Action (VLA) models alongside the efficiency and adaptability of specialist policies. Our proposed diffusion transformer-based specialist can achieve finer-grained control, efficiently incorporate any sensory input, and adapt to diverse tasks and heterogeneous environments with minimal data and training costs. By fostering synergistic cooperation, RoboDual effectively addresses several limitations inherent in existing VLAs, offering a cost-effective and widely adaptable solution for the practical deployment of large generalist policies.

**Limitations and future work.** In RoboDual, we assume that the inference time for both the generalist and specialist remains constant. Consequently, during deployment, each generalist inference step is associated with a fixed number of specialist steps. Besides, current generalist outputs discretized actions only. Given the scalability of auto-regressive VLAs, there is potential for the generalist to be trained to perform task decomposition (Ahn et al., 2022), affordance grounding (Qian et al., 2024; Lai et al., 2024), and image goal generation (Sun et al., 2024). These capabilities may serve as effective complements, establishing better "bridges" between the generalist and specialist models.

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

*Appendix*

## A    EXTENDED DETAILS ON EVALUATION SUITES

**CALVIN Benchmark.** CALVIN (Mees et al., 2022b) encompasses 34 distinct tasks, characterized by unconstrained task instructions that cover a range of skills, from basic pick-and-place operations to articulated object manipulation. The benchmark comprises four different environments as shown in Figure 7, each featuring a Franka Panda robotic arm for tabletop manipulation. In our study, we adopt the challenging evaluation setting, where policies are trained with demonstrations from environments A, B, and C, followed by zero-shot evaluations in environment D. The evaluation protocol includes a test set of 1,000 unique instruction chains, each composed of five consecutive tasks. The results of the baselines are directly referenced from the official benchmark. For implementation on CALVIN, we train the generalist model with LoRA (Hu et al., 2022) for 50 epochs. The specialist model is trained for 100k iterations with a batch size of 64 ($\sim$8 epochs). We set the action chunk size to $k_g = k_s = 8$ for both the generalist and specialist policy. Unless specified otherwise, our specialist model takes as input RGB and depth images from static (third-view) and gripper view cameras. Both the generalist and specialist operate on images of size $224 \times 224$.

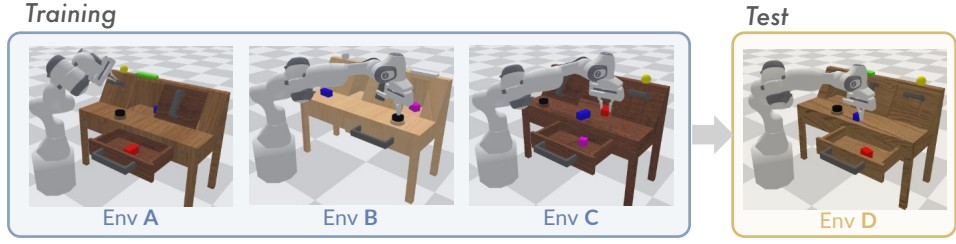

Figure 7: **Experiment setting of CALVIN.** CALVIN consists of four simulated environments (designated as A, B, C, and D), which differ in textures and object positions.

**Real-world experiments.** We propose an array of tasks to evaluate different manipulation skills, such as basic pick and place ("put the block into bowl"), non-prehensile manipulation ("push block left"), and articulated object manipulation ("open the drawer"), targeting at a comprehensive assessment of different policies. Due to limitations of our real-world setup, both the generalist and specialist take as input RGB images from a single camera view, with size $192 \times 256$. Notably, all generalist baselines (*i.e.,* Octo (Ghosh et al., 2024) and OpenVLA (Kim et al., 2024)) show zero success rate if we directly deploy them in a zero-shot manner. In addition to the challenges posed by novel camera views and embodiment configurations, current generalist policies face limitations in direct deployment due to being trained in normalized action space. The statistics of normalization are tailored to each dataset in the OpenX collection (Padalkar et al., 2024), resulting in an implicit, dataset-specific mapping between action spaces and corresponding observations. The reasons mentioned above raise the necessity of further adapting generalist policies with our self-collected demonstrations. We train the specialist baselines from scratch, while the generalists benefit from initialization with pre-trained checkpoints. All models undergo extensive training to ensure clear convergence over time.

The implementation of RoboDual follows a two-stage training pipeline, where the generalist is trained with multi-task data using LoRA (Hu et al., 2022) finetuning, followed by the efficient training of our specialist model with either multi-task or task-specific demonstrations. We set the action chunk size $k_s = 8$ for the specialist and use temporal aggregation during inference to achieve smoother control. Specifically, action outputs are temporally aggregated with an exponential weighting scheme $w_i = \exp(-m \cdot i)$, where $i \in [0, k_s]$ and $w_0$ is the weight for the oldest action. We set $m$ to 0.1 by default. DDIM scheduler (Song et al., 2021) is employed with the timestamps set to 100 for training and only 5 for inference. Directly predicting samples shows to be more robust than noise prediction. We find that increasing the denoising steps does not necessarily bring performance improvement, while using smaller steps allows faster inference and more responsive control. Classifier-free guidance (Ho & Salimans, 2022) is applied on generalist latents and proprioceptive states with a guidance scale of $w_g = 3$ for better controllability. We also explore the capability of the generalist to anticipate and plan ahead, generating actions $k_s$ steps

beyond its current observation. Consequently, we observe that the specialist is able to accurately "interpolate" actions based on conditioning from the generalist, guiding the robotic arm to the intended position and enabling precise, fine-grained control.

## B ARCHITECTURE DESIGN AND TRAINING HYPERPARAMETERS

**Generalist.** The architecture of our generalist policy is identical to Prismatic-7B (Karamcheti et al., 2024). Regarding training of the generalist, we follow OpenVLA (Kim et al., 2024), and use the AdamW optimizer with a constant learning rate of 2e-5 and weight decay of 0.01. We find that achieving robust convergence on the CALVIN benchmark, with its 34 distinct tasks and diverse environments, necessitates a sufficiently large batch size of 2048. We leverage gradient accumulation to achieve a large batch size with constrained compute. However, for real-world experiments, a smaller batch size of 64 remains feasible, offering greater flexibility in practical applications.

**Specialist.** We provide the detailed architecture information of our specialist policy in Table 5. Rotary position embeddings are incorporated into each DiT block to ensure coherence among chunked actions. In general, it is more lightweight than previous popular specialist policies (Zhao et al., 2023; Chi et al., 2023) that contains 50M to 80M parameters. Incorporating a new sensory input entails an individual ViT encoder (6.4M) and a perceiver resampler module (1.1M), which introduces additional 7.5M parameters in total. Considering an input size of 224, sensory encoders generate 196 tokens with a patch size of 16. Observation embeddings are subsequently downsampled to only 8 tokens by the resampler. Notably, conditioning latents in the cross-attention module only need to be computed once for the multistep denoising process, leading to more efficient control.

During training, we leverage the AdamW optimizer with a learning rate of 1e-4 and weight decay of 1e-3, mostly following Chi et al. (2023). A cosine-annealing scheduler with warm-up steps of 1,000 is employed to improve training stability. We apply classifier-free guidance (Ho & Salimans, 2022) on generalist latents and proprioceptive states with a condition drop chance of 0.1, allowing the specialist policy to imitate without privileged information from the generalist. No specific modifications of training parameters are made for any tasks or environments.

Table 5: **Architecture details of our diffusion transformer-based specialist policy.**

| Architecture of Specialist Policy | | |
|---|---|---|
| Diffusion Transformer | Layers | 6 |
| | Heads | 8 |
| | Hidden Size | 256 |
| | MLP Ratio | 4 |
| | Action Steps | 8 |
| Perceiver Resampler | Layers | 1 |
| | No. Tokens | 8 |
| | MLP Ratio | 4 |
| Sensory Encoders (ViT) | Layers | 6 |
| | Heads | 8 |
| | Hidden Size | 256 |
| | MLP Ratio | 4 |
| | Patch Size | 16 |
| Parameters | | 16.2M |

## C EXTENDED EVALUATIONS ON GENERALIZABILITY

We conduct extended the generalization experiment with multiple distractors at varied locations, as illustrated in Figure 8. We still pick "Put Block into Bowl" as the task for evaluation, but the environment is different from what is introduced in Figure 4. Beyond visual distractions, the block is placed at randomized positions to also evaluate position generalizability. Due to hardware constraints, the following experiments are conducted with a NVIDIA RTX 4060 laptop GPU with only 8GB memories. We perform 4-bit quantization to OpenVLA and our generalist model to fit

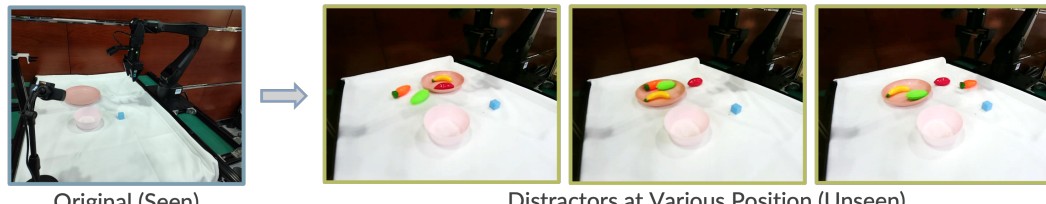

Original (Seen)    Distractors at Various Position (Unseen)

Figure 8: **Experiment setting for our extended generalization evaluation.** We place multiple different objects at various positions in the scene.

| Method | Success Rate |
|---|---|
| Diffusion Policy | 26.7 |
| OpenVLA | 46.6 |
| RoboDual | **60.0** |

Table 6: **Extended generalization evaluation.** RoboDual shows robustness under diverse distractions and object positions, achieving better performance over specialist and generalist-only baselines.

in the device. Specialist of RoboDual can still run at full precision. Experiment results are given in Table 6. RoboDual demonstrates superior generalizability over Diffusion Policy and OpenVLA, two representative specialist and generalist-only policies. We also explore whether RoboDual can generalize from "blue blocks" to "carrots" and achieve robust manipulation with video playing (dynamic visual changes) in the background. We have uploaded corresponding video demos to our anonymous project page.

## D  FAILURE ANALYSIS

During the 100 testing runs of the all tasks in our real-world environment, we record the causes behind each failure. Subsequently, we present these failure causes in a Sankey diagram, exemplified in Figure 9. We found one major failure issue would be "Not Following Instruction". The instruction-following ability of the VLA (generalist) model may not be fully leveraged by the specialist model to perform the desired task. It's worth future exploration of building better "bridges" beyond what is discussed in RoboDual (*i.e.*, discretized actions and generalist latent features) to facilitate a more synergistic framework.

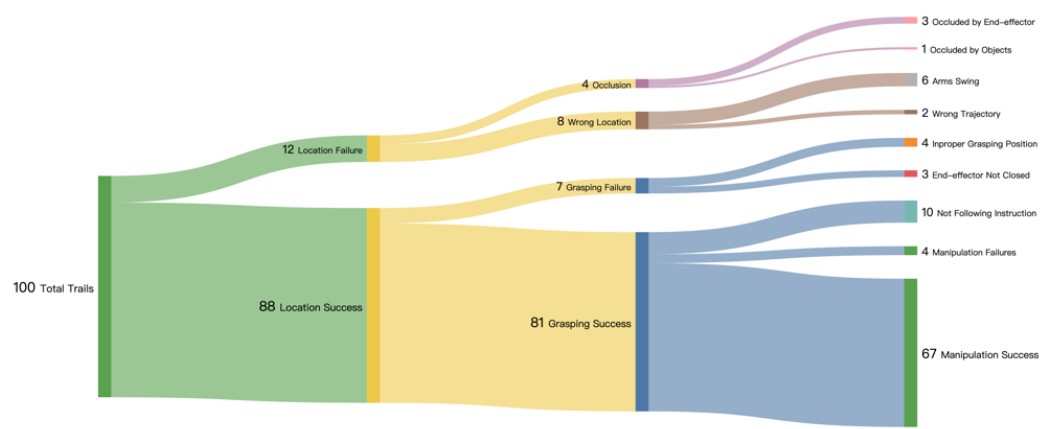

Figure 9: **Detailed failure case analysis on real-world robot experiments.**

