# OpenReview forum: "Towards Synergistic, Generalized, and Efficient Dual-System for Robotic Manipulation"
_ICLR.cc/2025/Conference — Submitted to ICLR 2025_

### Official Review · Reviewer_Sftm · 2024-11-02

**Soundness:** 3
**Presentation:** 2
**Contribution:** 3
**Rating:** 6
**Confidence:** 4

**Summary:**

To enable efficient fine-tuning and inference of VLA models, while not compromising generalisability, the article presents RoboDual, a dual-system approach that combines generalist and specialist policies to enhance robotic performance in dynamic environments. The generalist offers adaptability, while the specialist ensures efficient task execution. This synergy results in significant improvements in both real-world and benchmark tasks, offering strong performance with minimal data and higher control efficiency.

**Strengths:**

- The figures in the paper are thoughtfully designed and highly informative, significantly aiding readers in understanding the proposed methods and results.

- The dual-system approach aligns well with principles from cognitive science, making its application to embodied manipulation both insightful and innovative. Implementing this concept in robotics is a valuable contribution to the field.

- The extensive experimental results provide strong evidence of the model's advantages in achieving generalizable real-world manipulation. *RoboDual* outperforms both generalist and specialist-only models, demonstrating superior training and data efficiency, which highlights its practical value for broader real-world applications.

**Weaknesses:**

- When considering the system from another perspective: viewing the generalist as a vision-language model (VLM) and the specialist model as an action head, RoboDual can be seen as an asynchronous variant of Octo (Ghosh et al., 2024). This diminishes the novelty of the proposed approach, as the heterogeneity between the two systems mainly lies in the data and model scale, which impacts generalizability. It raises the question of whether scaling up the training data for the specialist model might yield comparable performance to the RoboDual system in terms of both computational efficiency and generalizability. Given that DiT is a scalable architecture, and considering the limited dataset used for experiments in Figure 5 (CALVIN), it would be valuable to explore this.


    One way to better distinguish these two systems is to draw more deeply on cognitive science concepts, such as viewing one system as responsible for reasoning (akin to System 1) and the other for actuation (akin to System 2). For example, in a task like "write down the answer of 1 + 1 on the blackboard," the reasoning required to determine the answer is challenging for the specialist system alone. Highlighting such distinct roles could provide a more fundamental differentiation between the two systems.

> Dibya Ghosh, Homer Walke, Karl Pertsch, Kevin Black, Oier Mees, Sudeep Dasari, Joey Hejna, Tobias Kreiman, Charles Xu, et al. Octo: An open-source generalist robot policy. arXiv preprint arXiv:2405.12213, 2024.

- The experiments on training efficiency could be improved. The use of DistillBERT might not be sufficient for capturing the semantics of actions and target objects from language instructions. I would suggest adding two additional baselines:

    1. Using T5 to encode language for the specialist-only model to ensure sufficient semantic extraction. Since language encoding is performed once per rollout, T5-xxl might be a suitable choice.

    2. GPU hours may not be the best metric for measuring efficiency, as it does not account for the number of parameters. I recommend switching the x-axis metric to FLOPs for a more accurate representation of computational efficiency.

**Questions:**

- **Line 26**: Could you clarify why "with" is italicized?

- **Line 268**: The description of the shifted-window conditioning mechanism is somewhat unclear. Why are only $k_g - \tau_s$ generalist actions sampled as conditioning rather than using the entire chunk of $k_g$ actions?

- **Line 197**: There appears to be a duplicated closing parenthesis in ")), \etc". Could you confirm if this is an error?

- In the experiment described in Figure 5, is the generalist model in the dual approach frozen? If it is frozen, are the weights solely from OpenVLA, or has it been further fine-tuned on CALVIN?

- Is the VLA model strictly necessary as the generalist model? If a vision-language model (VLM) were used to extract conditions instead, would this achieve comparable performance to RoboDual?

---

> ### Author Response · Authors · 2024-11-21
> **Authors' Response to Reviewer Sftm**
>
> Thanks for your careful review and valuable comments. We address each question below.
>
> > *${\color{BrickRed}W1:}$* (1) RoboDual can be seen as an asynchronous variant of Octo. (2) The heterogeneity between the two systems mainly lies in the data and model scale, which impacts generalizability. (3) Whether scaling up the training data for the specialist model (DiT) might yield comparable performance to the RoboDual system in terms of both computational efficiency and generalizability.
>
> 1. **We believe RoboDual distinguishes itself from Octo in multiple aspects**.
>
>   a. **Architecture:** As the reviewer also recognized, we employ scalable DiT architecture with elaborated conditioning mechanisms as our specialist policy. Whereas in Octo, they employ an MLP decoder with a diffusion objective.  Our DiT-based specialist model can better model the temporal relation of consecutive actions with causal attention, and can be conditioned effectively with multi-source conditions through cross-attention mechanism. Its scalability would also be an interesting aspect for future exploration.
>
>   b. **How to achieve asynchronous execution:** The prerequisite of asynchronous execution of the two systems is that our specialist has its own observation encoders and the two systems can inherently run independently. The action decoder in Octo takes as input solely the transformer latents, thus it cannot be adapted to rapidly updated observation inputs with fixed transformer latents for asynchronous execution.
>
> 2. **In our humble opinion, the heterogeneity between the two systems in terms of data and model scale brings benefits that outweigh the potential drawbacks in generalization**.
>
>   a. It allows the VLA-based generalist to be pretrained on web-scale VQA data that the diffusion-based specialist policy cannot leverage.
>
>   b. The model scale gap is mostly motivated for achieving faster and smoother control through the efficient specialist model. Higher control frequency itself contributes to a non-negligible extent to the superior performance of RoboDual, as OpenVLA shows a low success rate on every task that needs certain dexterity.
>
>   c. We acknowledge the reviewer's concern that the generalizability of the generalist may not be fully exploited by the specialist. In our experiments, the robustness of free-form language instructions (Table 2) and diverse visual variations (Table 3) validate the generalizability of RoboDual.  We upload video demos of further generalization experiments to the anonymous project page.
>
> 3. **Scalability of DiT**: In fact, we tried to scale up the DiT-based specialist in CALVIN, but found minimal gains (3.69 with 130M parameters vs. 3.66 with 17M parameters). One possible reason, as the reviewer mentioned, is the limited dataset size. In addition, for real-world experiments, scaling up the specialist's model size can inevitably diminish our boost to the control frequency, which does not necessarily translate into improved performance on certain tasks. As for data scalability, we show how data efficient the specialist is within our RoboDual framework with results listed in Table 4(a). Overall, we acknowledge the reviewer's valuable feedback and are actively exploring this direction in our extended work of RoboDual.
>
> > *${\color{BrickRed}W2:}$*  Distinguish these two systems with cognitive science concepts.
>
> Thanks for your constructive feedback on highlighting our dual-system synergy from the perspective of cognitive science. Performing higher-level reasoning tasks highly depends on the capabilities of generalist policy. We tried to highlight the distinctness between the two systems with the following experiments: (1) Following OpenVLA, we show that RoboDual can excel in multi-instruction tasks where specialist policies generally struggle with (Figure 3), where the slow-system (VLA) assist with semantic understanding. (2) In the data efficiency experiment, the specialist can effectively extrapolate to new tested positions not included in 5 training samples, thanks to conditioning information from the generalist.
>
> Tasks such as "write down the answer to 1 + 1 on the blackboard" require mathematical and advanced reasoning abilities, which is popular for LLM research while presenting significant challenges for current VLA models. This also falls outside the scope of the pretraining dataset (OpenX). Thanks for the suggestion and we will investigate this in our future research.

---

> ### Author Response · Authors · 2024-11-21
> **Authors' Response to Reviewer Sftm (continued)**
>
> > *${\color{BrickRed}W3:}$* The experiments on training efficiency could be improved. (1) Using T5 to encode language for the specialist-only model to ensure sufficient semantic extraction. (2) FLOPs could be a better metric than GPU hours to measure the training efficiency.
>
> (1) Thanks. As suggested, we have added the experiment results using T5-xxl as the language encoder for our specialist-only model and updated Figure 5 correspondingly. We found that switching to larger language encoders only brings marginal performance improvement on CALVIN (from 0.45 to 0.53 after 100k iterations of training). We believe the limited capacity of our specialist model is the main bottleneck. This result also highlights the merit of dual-system synergy and conditions provided by a VLA (generalist) model. Conditioning information from the generalist model is much more informative than merely language embeddings.
>
> (2) GPU-hours is an intuitive metric directly reflecting the computational resources consumed. Our advantage of training efficiency over the generalist-only variant will get more pronounced when using FLOPs as a metric, as the generalist is frozen when adapting the specialist model. The back-propagation of gradients on the generalist side is thus eliminated. We follow OpenVLA and popular LLMs (i.e., LLaMA-2) to only report GPU hours as the reflection of computation cost and carbon footprint.
>
>
>
> > *${\color{BrickRed}Q1:}$* Line 26: Could you clarify why "with" is italicized?
>
> We intended to highlight that the generalist and specialist are working together as a whole. We have revised it in the updated manuscript.
>
> > *${\color{BrickRed}Q2:}$* Line 268: Why are only generalist actions sampled as conditioning rather than using the entire chunk of actions?
>
> The shifted-window conditioning mechanism stems from the asynchronous execution of two systems. As discussed in Section 3.2, during inference, the slower generalist model may consistently 'lag behind' the faster specialist. Specifically, the specialist operating at timestamp $t+k$ must learn from action outputs produced by the generalist at timestamp $t$. To address this asynchronicity, we propose a sliding-window-based conditioning mechanism.
>
> > *${\color{BrickRed}Q3:}$* Line 197: There appears to be a duplicated closing parenthesis in ")), \etc". Could you confirm if this is an error?
>
> Thanks for the careful review! We have corrected the typo.
>
> > *${\color{BrickRed}Q4:}$* Figure 5: Is the generalist model in the dual approach frozen? Has it been further fine-tuned on CALVIN?
>
> Yes, the generalist is frozen while only the specialist is trainable within our approach. As mentioned in Lines 200-205, the generalist is finetuned on CALVIN considering the domain gap between OpenVLA's pretraining dataset (containing only real-world datasets) and CALVIN simulation. The original OpenVLA might generate unreasonable outputs and thus provide barely informative conditioning to the subsequent specialist, if directly deploying it in a zero-shot manner. We intend to take experiments on CALVIN as an artifact and help the community better reproduce our results. We believe it's possible to perform zero-shot adoption of OpenVLA, along with the efficient tuning of specialist, to achieve competitive results only if the evaluation setup is within the scope of Open X-Embodiment pretraining (*e.g.*, using WindowX Robot as in Bridgev2).
>
> > *${\color{BrickRed}Q5:}$* Is the VLA model strictly necessary as the generalist model? If a vision-language model (VLM) were used to extract conditions instead, would this achieve comparable performance to RoboDual?
>
> Thanks for the insightful question. The framework of RoboDual is applicable to VLM-based generalist, while it might be hard for it to catch up with VLA-based model. In our preliminary experiments, we tried to directly leverage Prismatic-7B VLM as the generalist model. In CALVIN experiments, the performance is just slightly higher than the specialist-only variant (0.45 average length). We analyze the results as follows:
> - The pretraining of VLA models on large-scale robot in-domain data (e.g., Open X-Embodiment) is greatly beneficial for robotic manipulation tasks and for the adaptation of specialist models in RoboDual.
> - Existing visual language models (VLMs) are primarily trained for high-level scene understanding tasks, such as visual question answering (VQA). Consequently, they struggle to capture the temporal evolution of the roll-out process and tend to generate only global description latents for conditioning. As a result, they offer less informative conditions for the sequential decision-making process when compared to visual language action (VLA) models.
> - Our practices also align with prior research (*e.g.*, RoboFlamingo, LCB) that incorporates VLMs into manipulation policies, wherein the VLMs are further fine-tuned using robotic data.

---

> > ### Comment · Reviewer_Sftm · 2024-11-25
> > **Thank you for your prompt reply**
> >
> > Thank you for the detailed explanations and the effort you’ve put into addressing my concerns. These discussions address most of my points, though I still have some reservations about specific aspects of the response:
> >
> > - **Reply to W3:** I’m not entirely convinced about leaving out the computational overhead of the frozen modules. This approach implies that the FLOPs for fine-tuning the specialist could be high, as it requires inference with the 7B generalist. Considering these factors might provide a more comprehensive and transparent evaluation of efficiency.
> > - **Reply to Q2:** I now understand the mechanism behind the asynchronous conditioning. However, I find the description in the paper somewhat vague. Enhancing the clarity of this section would significantly improve the paper’s readability and comprehension.
> >
> > Besides, I found your reply to Q5 particularly insightful and well-articulated. Based on the overall improvements and the effort demonstrated, I have decided to raise my score.

---

> > > ### Author Response · Authors · 2024-11-25
> > > **Thanks for your prompt discussion and recognition of our efforts**
> > >
> > > Thanks for considering our responses and recommending acceptance. We will update our paper regarding the efficiency analysis and asynchronous conditioning mechanism to further improve clarity.

---

### Official Review · Reviewer_AWwV · 2024-11-03

**Soundness:** 3
**Presentation:** 3
**Contribution:** 3
**Rating:** 3
**Confidence:** 4

**Summary:**

The paper introduces RoboDual, a dual-system framework combining generalist and specialist policies for robotic manipulation. RoboDual leverages both i) a generalist’s broad generalization capabilities with ii) a specialist’s task-specific precision and efficiency. The generalist is a large-scale pretrained vision-language-action model which provides high-level guidance, while the specialist is a diffusion policy which facilitates rapid, precise control.

**Strengths:**

1. Combining a high-level generalist and a low-level specialist model is a compelling paradigm to enable broader generalization while maintaining more fine-grained control.

2. RoboDual achieves higher performance with fewer demonstrations and limited computational requirements.

**Weaknesses:**

1. A bi-level policy increases model complexity and therefore inference time, which may affect performance in low-latency tasks.

**Questions:**

1. How does the performance vary with different sensory input combinations, and could simpler setups still achieve competitive results while offering advantages in runtime efficiency?

2. How well does RoboDual perform in more dynamic or even user-interactive settings (e.g. moving an object while a trajectory is being executed)?

---

> ### Author Response · Authors · 2024-11-21
> **Authors' Response to Reviewer AWwV**
>
> Thanks for your valuable review. We address your concerns below.
>
> > *${\color{BrickRed}W1:}$* A bi-level policy increases model complexity and therefore inference time, which may affect performance in low-latency tasks.
>
> As highlighted in our paper and videos from our anonymous project page, the generalist and specialist in our bi-level policy are asynchronously executed (Line 266), which, on the contrary, improves performance in low-latency tasks. To be more specific, considering the inference latency of generalist and specialist to be $T_{g}$ and $T_{s}$ respectively, asynchronous execution allows us to perform $k$-step actions with only **one** generalist inference paired with $k$ steps of specialist inference. The resulting inference time would be $T_{g} + k T_{s}$, where employing the generalist solely requires  $k T_{g}$. Given that our specialist is highly efficient ($T_{s} \ll T_{g}$) with only ~17M parameters (excluding the vision encoder), RoboDual will be more efficient than generalist-only policies as long as $k\geq 2$. In practice, we use $k=8$.
>
> > *${\color{BrickRed}Q1:}$* How does the performance vary with different sensory input combinations, and could simpler setups still achieve competitive results while offering advantages in runtime efficiency?
>
> We did relevant ablation studies shown in Figure 6(b), showcasing how our specialist policy can leverage additional sensory inputs beyond the third-view RGB to further boost performance. Since it would take a huge burden for us to iterate over all possible combinations, we hope the current experiments are informative.
> As indicated in Figure 6(b), using only static (third-view) RGB input yields a competitive result of 3.54 average length on CALVIN, which is still superior to 3D Diffuser Actor that leverages static and gripper-view depth inputs with camera parameters (as shown in Table 1).
>
> > *${\color{BrickRed}Q2:}$* How well does RoboDual perform in more dynamic or even user-interactive settings (e.g. moving an object while a trajectory is being executed)?
>
> Thanks for the question. During the rebuttal, we conduct additional tests and show video demos on our anonymous project page, where RoboDual is able to recover from failure and try to regrasp the block when the first attempt is missed. This case shows the generalizability of RoboDual under unprecedented dynamic scenarios where the block is dropped in an uncontrolled position and pose. Note that such cases are out of the training distribution. To further address your concern, we add additional experiments with user-interactive settings, where we introduce intentional human interference during the roll-out process of RoboDual. Corresponding experiment videos are uploaded to our anonymous project page (see "More Generalization Experiments" section).

---

> > ### Comment · Reviewer_AWwV · 2024-12-02
> >
> > I thank the authors for their response. My main concern was the performance of this method in the context of more dynamic tasks which require lower latency in planning/re-planning. In general, this concern still remains. In particular,
> >
> > 1. Regardless of whether or not the framework facilitates asynchronous execution of the generalist and specialist policies, the fact of the matter is in low-latency contexts, more frequent re-planning is required. Thus, even if the generalist and specialist are asynchronously run, lower latency places a greater burden on more frequent communication (or synchronization) of their contexts. That is, while execution time might be lower for more static tasks, the __effective__ execution time would be much higher for more dynamic tasks. The degree to which this is a problem has not been investigated in the current iteration of this work.
> >
> > 2. I remain unconvinced by the regrasping example the authors provide as evidence for execution in dynamic tasks. In particular, this task is not actually dynamic --- it can simply be viewed as a "stitching" of two static pick-and-place style trajectories, one after another.
> >
> > In summary, my final rating is in favor of rejection. In any work that proposes a dual system (e.g. combining a high-level generalist and a low-level specialist model), issues of execution time/latency are absolutely critical. I recommend the authors address this axis more explicitly, especially in investigating the utility of the proposed framework under dynamic tasks closer to real-world settings.

---

> ### Author Response · Authors · 2024-12-02
> **Looking forward to your prompt response**
>
> We sincerely hope that we have addressed all of your concerns satisfactorily. **As the rebuttal phase is about to conclude**, we would greatly appreciate your prompt response. Please feel free to share any further comments or concerns you may have.

---

> ### Author Response · Authors · 2024-12-03
>
> With all due respect, we acknowledge the reviewer’s reply post close to the end of discussion given the initial brief comment. The core concern is "dynamic tasks which require lower latency in planning".
>
> ### **Latency**
>
> Concretely, in the initial review, the reviewer questions the **overall inference time** concerning low-latency planning. In the latest comment, from our understanding, the reviewer questions the **communication latency** between the generalist model and specialist model. We address the questions below.
>
> We totally agree with reviewer’s claim that a robotic system requires efficient task execution under dynamic tasks. This requirement applies to each algorithm, *whether or not* it is a dual-system. However, many prior works addressing bi-level policies [1,2,3] fail to fully consider this perspective.
>
> [1] Ahn, Michael, et al. "Do as I can, not as I say: Grounding language in robotic affordances." CoRL (2022).
> [2] Driess, Danny, et al. "PaLM-E: An embodied multimodal language model." ICML (2023).
> [3] Li, Xinghang, et al. "Vision-language foundation models as effective robot imitators." ICLR (2024).
>
> *RoboDual improves upon these works and generalist-only baselines, as highlighted throughout the paper and also recognized by Reviewer Sftm and bxCB*.
> 1. We improve the **overall control frequency** from 4Hz (OpenVLA only) to 15Hz, as highlighted in Line 475-486 in the paper and mentioned explicitly by Reviewer bxCB. Besides, the communication between two systems entails task latents ($\mathbb{R}^{32 \times 256}$), action latents ($\mathbb{R}^{7 \times 256}$), and discretized actions ($\mathbb{R}^{7}$). The total information volume is around 39.9 KB (FP32), and the **communication latency** with shared memory or pipes is in the range of microseconds to a few milliseconds. Given a single communication can support multi-step reasoning of the specialist, the burden on communication is also negligible.
> 2. Based on the above discussion, we'd like to clarify that **the bottleneck of inference latency lies in the large generalist model, instead of specific designs involved in our dual-system architecture**. While using a smaller generalist model or specific engineering techniques could further mitigate the latency bottleneck, this is beyond the scope of our current work.
>
> ### **Dynamic Tasks**
>
> We would appreciate it if the reviewer could have provided explicit experimental settings so we could conduct further experiments concerning dynamic tasks in the initial review. By far, we would like to discuss it from the following aspects:
> 1. On our anonymous project page, we show that RoboDual can successfully perform dynamic tasks under: (1) Unpredited (Unseen) dynamics with the grasped object is dropped uncontrollably; (2) Dynamics introduced by actively interfering with the position of an object with the human hand during execution; (3) Background dynamics with a random video playing in the scene.
> 2. We hypothesize the reviewer is referring to the dynamics associated with continuous and unpredictable motion of objects to be manipulated. We kindly note that the majority of current literature on robotic manipulation does not address such scenarios, including our specialist and generalist baselines (e.g., Diffusion Policy, ACT, Octo, and OpenVLA) and most related bi-level policies (e.g., PaLM-E, SayCan, RoboFlamingo, etc). **They all focus on quasi-static situations**. Moreover, extremely dynamic cases are not, nor should they be, within the scope of RoboDual's objectives. We believe the unique contributions of RoboDual could not be diminished.
>
> We would keep continuing polishing the work as future work as suggested. Thanks.

---

### Official Review · Reviewer_bxCB · 2024-11-03

**Soundness:** 3
**Presentation:** 3
**Contribution:** 2
**Rating:** 6
**Confidence:** 3

**Summary:**

This paper introduces a new method for solving language-conditioned, robotic manipulation tasks.  The proposed method, RoboDual, combines an vision-language-action (VLA) model for high-level task understanding and long-horizon reasoning with a low-level policy to handle spatial reasoning and precise movements.  The two models are integrated together by passing discretized action predictions and latent representations from the generalist model to the specialist model.  A key benefit of their approach is the ability to run at higher control frequencies (20Hz), which is necessary for many dynamic manipulation skills, since they do not rely on the generalist to make predictions at every time step.  RoboDual is evaluated on a suite of challenging manipulation tasks in simulation and the real-world, where it outperforms all baselines in terms of success rate and shows strong robustness to across task variations.  RoboDual also outperforms baselines even in settings where the amount of data is significantly reduced.

**Strengths:**

- The paper proposes a scheme for combining a generalist VLA model with a specialist low-level policy model, via conditioning on latent representations and discretized actions.  This approach relies on the low-level policy to process non-vision-language inputs (like depth, proprioceptive, or tactile info), so the VLA does not need to be fine-tuned extensively.
- RoboDual can be trained much faster than a fully generalist approach, since the action predictions of an under-trained VLA model are refined by the specialist model that trains quickly.
- The experiments in Section 4.4 show that RoboDual is very sample efficient, achieving 73.3% success rate at 5 demos on real-world tasks.  This indicates that the coarse action predictions of the generalist are helpful and enable the specialist to refine the actions with limited data.-
- RoboDual can be run at 15Hz at inference, compared to 4Hz for openVLA.  This difference is significant, since jumpy movements of the robot prevent it from solving tasks that require precision.

**Weaknesses:**

- The ablation study needs some work.  Please switch to a different color map so that it is easier to distinguish the bars and read the legend.   The axis range makes it look like the ablations have a substantial impact on model performance, even though the difference in performance is minimal.  There should be error bars, otherwise it is difficult to determine whether a 0.03 increase in average time is significant.  The discussion of these results should also be changed to better reflect the actual results.  For instance, it is not true that "each conditioning source from the generalist model plays an **essential** role in ... enhancing overall performance" [emphasis mine] if removing the conditioning decreases average length by at most 4%.
- There are some instances where the wording could be improved.   In Figure 1 caption: "the fast specialist policy *obsesses* ..." (achieves?). Top of page 2, "The *yielding* policy" (The resulting policy).  Bottom of page 2, "We bring in a novel approach" (We introduce? a novel approach).  Beginning of Section 3.3, "Disparate from" (Unlike?).
- The first contribution says, "Our methodology ... paves the way for the practical application of VLA models".  This is quite a broad claim.  I believe you are hinting at the computational efficiency of the dual-system.  Perhaps modify this to say "practical application of VLA models to higher-frequency control tasks".

**Questions:**

- In Table 3, it is interesting that transferring to an unseen background (checkered to solid-white tablecloth) results in 30% or greater drop in performance for all models.  Do you have a hypothesis for why this is the case?  One would expect the generalist models and RoboDual to be more robust to background texture.
- What was the reason for choosing joint space control over end-effector control?
- In Section 3.3, it says that the specialist model is trained for "one hour" but in Section 4.4 it says "one hour of training on a node equipped with eight A100 GPUs".  Is this the same "one hour"?  If so, updating Section 3.3 to "8 gpu-hours" would be more accurate.
- It seems that fine-tuning the generalist policy to predict discretized actions in a specific robot's action space makes it no longer "generalist".  Have you thought of other ways to condition the low-level policy that might allow one to deploy the system on different robot types?

Typos: Figure 6a legend: (w/o Action L**e**tent).

---

> ### Author Response · Authors · 2024-11-21
> **Authors' Response to Reviewer bxCB**
>
> Thanks for your careful review and we really appreciate your comments. We address your questions below.
>
> > *${\color{BrickRed}W1:}$*  The ablation study needs some work. (1) Switch to a different color map so that it is easier to distinguish. (2) There should be error bars. (3) The discussion of these results should also be changed to better reflect the actual numerical results.
>
> Thanks for the advice. We have revised the figure and the results discussion part as suggested. Please see the updated manuscript for the modifications.
>
> > *${\color{BrickRed}W2:}$*  There are some instances where the wording could be improved.
>
> Thanks. We have revised our paper accordingly and improved the overall presentation clarity.
>
> > *${\color{BrickRed}W3:}$*  The first contribution is a broad claim. modify this to say "practical application of VLA models to higher-frequency control tasks".
>
> Agreed. We have modified the contribution claim with more highlights on the high frequency in terms of practical application. However, we would also like to emphasize that enhancements in overall performance and training efficiency are also critical factors contributing to the successful real-world deployment of VLAs.
>
> > *${\color{BrickRed}Q1:}$*  In Table 3, it is interesting that transferring to an unseen background results in 30% or greater drop in performance for all models. Why?
>
> One possible reason is the reflective surface of the leather texture of the solid-white tablecloth. We place the block in three different positions and find our baselines can only succeed in one specific position (33.3% success rates and lower for our baselines as illustrated in Figure 3). We also put rigorous evaluation criteria on this task, where only smoothly pushing the block towards the left by 5cm is deemed as a success. The dexterity highlighted in the non-prehensile manipulation task regarding pushing a small and light block also adds difficulty and leads to a lower success rate of all methods, even without background change. We would also like to clarify that OpenVLA and RoboDual do show greater generalizability, where the success rate of OpenVLA remains 20% as in the original background, and the relative performance decline is 18% for RoboDual and 100% for ACT, respectively. The limited model capacity of Octo (83M parameters) also hinders its robustness under this setting.
>
> > *${\color{BrickRed}Q2:}$*  What was the reason for choosing joint space control over end-effector control?
>
> We leverage the 7-DoF end-effector action space with end-effector position (3), orientation (3), and the gripper state (1) in both CALVIN simulation and Real-world experiments. We will revise the paper to make it clear.
>
>  > *${\color{BrickRed}Q3:}$*   Inconsistency of expression in Sections 3.3 and Section 4.4.
>
> Thanks for your careful review. We've revised the paper with  "8 GPU-hours" in Section 3.3 for better coherency.
>
> > *${\color{BrickRed}Q4:}$*  Any other ways to condition the low-level policy that might allow one to deploy the system on different robot types?
>
> Thanks for your insightful feedback. End-to-end VLA methods (RT-2 and OpenVLA) directly output the normalized low-level action, thus they have to be fine-tuned in new environments or embodiments with heterogeneous action spaces. Same for our real-robot setting, which is sadly not included in Open X-Embodiment pretraining. However, as also mentioned in the *future work* section of our paper, our dual-system synergy can be further extended to embodiment-agnostic generalist policies outputting higher-level action abstractions like point flow (e.g., ATM [1]) or latent action (e.g., LAPA [2]). We believe this is a promising direction to develop more advanced generalist policy, while also allowing data and training efficient adaptation on specific embodiments with RoboDual.
>
> Typos are fixed in the revised version and we have thoroughly inspected the manuscript.
>
> [1] Wen C, Lin X, So J, et al. Any-point trajectory modeling for policy learning. arXiv:2401.00025, 2023.
> [2] Ye S, Jang J, Jeon B, et al. Latent Action Pretraining From Videos. arXiv:2410.11758, 2024.

---

> > ### Comment · Reviewer_bxCB · 2024-11-26
> >
> > Thank you for the detailed response.  You have addressed all of my concerns and questions.

---

### Official Review · Reviewer_woVN · 2024-11-04

**Soundness:** 3
**Presentation:** 3
**Contribution:** 3
**Rating:** 6
**Confidence:** 4

**Summary:**

This paper investigates a pertinent question in imitation learning: how to combine the generalization capability of models like OpenVLA with the accuracy and task-specific precision of methods such as ACT or Diffusion Policy. To address this, the authors propose a novel framework, RoboDual, which uses the intermediate tokens in OpenVLA to condition a modified Diffusion Policy. Through simulation and real-world experiments, the proposed method demonstrates substantial performance improvements over both generalist and specialist baselines.

**Strengths:**

- **Effective Synergy**: The RoboDual framework innovatively combines a generalist and specialist model, leveraging OpenVLA’s generalization and Diffusion Policy's efficiency. This approach addresses a gap in imitation learning by merging generalist adaptability with specialist precision.
- **Experimental Results**: Both simulation and real-world experiments show significant performance gains over state-of-the-art baselines in generalization and task-specific adaptation, highlighting RoboDual's potential in practical settings.
- **Well-Written and Accessible**: The paper is clear, well-organized, and easy to understand, making the novel approach and its implications accessible.
- **Open-Source Commitment**: The authors promise to release the code, which could foster further research and replication.

**Weaknesses:**

1. **Limited Real-World Experiments on Generalization** Although the framework shows promise, its real-world experiments, particularly those evaluating generalization capabilities, remain limited. Conducting additional experiments—such as testing with a wider variety of novel objects (beyond just banana to eggplant), introducing more distractors at varied locations, using diverse language instruction templates, varying lighting conditions, or providing more detailed descriptions of the existing experiments—would significantly bolster the case for RoboDual’s practical applicability.

2. **Insufficient Introduction to the CALVIN Dataset** Including illustrative images in Section 4.2 to showcase the training and test settings of the CALVIN dataset would enhance readers' understanding of the experiment RoboDual run in simulation.

3. **Improved Color Differentiation in Bar Charts** The colors representing Octo, OpenVLA, and Ours (single/multi) in the bar figures are difficult to distinguish. Selecting more visually distinct colors would improve clarity and make comparisons easier.

4. **Failure Analysis** It is hard to tell which part is the bottleneck for the current method. A failure analysis will be helpful.

**Questions:**

### Questions
1. From the current experiments, this method does not seem to solve problems beyond what Robot-Moo or RoboPoint achieve. Any thoughts on how this method could outperform approaches based on explicit representations like bounding boxes or points?
2. It would be easier to understand if the conditioning feature were illustrated in Figure 2.
3. The authors claim that OpenVLA and Octo serve as generalist models, but they do not generalize effectively in all cases. For instance, the OpenVLA paper mentions challenges with out-of-distribution (OOD) cases, reflective surfaces, unseen action spaces, and actions along the depth axis. Given this, OpenVLA may not be an ideal generalist. Does this imply that RoboDual’s generalization is limited by OpenVLA’s capabilities?
4. In Figure 3, OpenVLA outperforms RoboDual in the "Knock down object" task. Can the authors explain why this is the case?
5. Additional real-world experiments on generalization ability would be beneficial. Including a baseline setting where Diffusion Policy/OpenVLA performs reasonably well would also help clarify RoboDual's improvements, given the claim that OpenVLA mainly provides generalization ability in this setup.
6. Perhaps I missed it, but is the Diffusion Policy baselines in your experiments the modified versions (specialist only), or the originals? This distinction is important, as a significant improvement from switching the transformer backbone to DiT may impact the novelty.
7. Is the OpenVLA in your experiments the same model used as the generalist in RoboDual (generalist only)?

---

> ### Author Response · Authors · 2024-11-21
> **Authors' Response to Reviewer woVN**
>
> Thanks for your detailed review. We address your questions below.
>
> > *${\color{BrickRed}W1:}$* Limited Real-World Experiments on Generalization.
>
> Thanks for the constructive feedback. We have studied the generalizability of diverse language instruction templates, as shown in Table 2. We also provide generalizability tests of four different axes in Table 3. RoboDual shows exceptional robustness compared to all specialist and generalist-only baselines. During rebuttal, we have added more generalization experiments introducing multiple distractors. Experiment settings are presented in Appendix C of the revised paper and we also upload new video demos to the Generalization Evaluation section of the anonymous project page.
>
> > *${\color{BrickRed}W2:}$* Insufficient Introduction to the CALVIN Dataset.
>
> Given the limited space of the main paper, we have added illustrative figures in Appendix A, paired with the introduction of our detailed experiment setting on CALVIN.
>
> > *${\color{BrickRed}W3:}$* Improved Color Differentiation in Bar Charts.
>
> Thanks for the suggestion. We have improved the readability of the bar plot by applying more visually distinctive colors.
>
> > *${\color{BrickRed}W4:}$* Failure Analysis.
>
> Agreed. We provide a detailed failure analysis with a Sanky plot in the updated Appendix D. In certain cases, we observe the instruction-following ability of the VLA (generalist) model may not be fully leveraged by the specialist model to perform the desired task. It's worth future exploration of building better "bridges" beyond what is discussed in RoboDual (*i.e.*, discretized actions and generalist latent features) to facilitate a more synergistic framework.
>
> > *${\color{BrickRed}Q1:}$* How RoboDual could outperform approaches based on explicit representations like bounding boxes or points?
>
> Thanks for the question. In this paper, we mainly focus on how to build a synergistic dual-system framework that leverages the broad generalizability of generalists and the efficiency and fast adaptation of specialists. As discussed in the *future work* section, it is feasible to enhance the existing generalist within RoboDual by incorporating the capability for affordance generation (explicit representations like points or bounding boxes). This enhancement has the potential to further improve the synergy between the two systems and optimize planning performance.
>
> Unlike the explicit representations (bounding boxes or points) used in Robot-Moo and RoboPoint, the VLA models in RoboDual provide high-level task understanding through latent tokens. This allows RoboDual to excel in multi-instruction tasks (Figure 3) and to maintain robustness against free-form language instructions (Table 2). The ablation study in Figure 6(a) also highlights the importance of latent representations.
>
> Furthermore, Robot-Moo and RoboPoint generate affordance proposals once solely at the beginning of execution for more efficient manipulation, hindering their adaptability to dynamic scenarios. RoboDual enables effective asynchronous synergy and allows for continuous updates for both high and low-level decision-making. Failure recovery demonstrations on our anonymous project page showcase RoboDual's adaptability to unpredicted changes.
>
> > *${\color{BrickRed}Q2:}$* It would be easier to understand if the conditioning feature were illustrated in Figure 2.
>
> Yes. In the context of sensory inputs and generalist latent variables utilized within the cross-attention module for conditioning, we have made efforts to align the color schemes of both the input and conditioning components in Figure 2. This alignment is intended to enhance readability and facilitate comprehension. We've increased the saturation of colors in the updated manuscript to enhance clarity.

---

> ### Author Response · Authors · 2024-11-21
> **Authors' Response to Reviewer woVN (continued)**
>
> > *${\color{BrickRed}Q3:}$* Does RoboDual’s generalization is limited by OpenVLA’s capabilities?
>
> Indeed, the semantic and high-level understanding ability of RoboDual could be mainly bottlenecked by OpenVLA and the Open X-Embodiment pretraining. It's possible to extend OpenVLA's pretraining scheme with web-scale VQA data, like RT-2, to achieve broader generalization. However, the diffusion-based specialist model more effectively captures the multimodality of actions and helps RoboDual perform better under tasks that need lower-level generalization, such as position variation.
>
> > *${\color{BrickRed}Q4:}$*  Why OpenVLA outperforms RoboDual in the "Knock down object" task?
>
> Thanks for the careful review. We would like to emphasize that the task designated as "Knock <obj> Over" necessitates the least degree of dexterity but highlights instruction-following ability. When testing RoboDual, we observed cases where the policy failed to adhere to the instructions, resulting in the incorrect object being knocked over. Though RoboDual shows notable performance improvement over specialist-only baselines equipped with T5 language encoders, the semantic understanding ability of our generalist model is not fully inherited and leveraged by the subsequent specialist policy. We have added the discussion in Appendix D.
>
> > *${\color{BrickRed}Q5:}$*  Additional real-world experiments on generalization. Include a baseline setting where Diffusion Policy/OpenVLA performs reasonably well.
>
> Thanks for the advice. We update the experiment results with multiple distractors at varied locations in Appendix C and list the results below. We also explore whether RoboDual can generalize from "blue blocks" to "carrots" and achieve robust manipulation with video playing (dynamic visual changes) in the background. We have uploaded corresponding video demos to our anonymous project page (in the Generalizability Evaluation section).
>
> Results with multiple distractors at varied locations (please refer to the updated manuscript for detailed experiment setting):
> |Methods|Success Rate|
> |-|-|
> |Diffusion Policy|26.7%|
> |OpenVLA|46.6%|
> |RoboDual (Ours)|*60.0%*|
>
> Since we adopt rigorous evaluation for our real-world experiments, it is hard to know the baselines' performance before designing the tasks. However, the task of "Lift the pod lid" could be an example to depict the gain when baseline methods perform relatively well. This task requires less generalization ability as the pod is placed in a fixed location.
>
> > *${\color{BrickRed}Q6:}$*  Is the Diffusion Policy baselines in your experiments the modified versions (specialist only), or the originals?
>
> We apply the best-performing variant as indicated in the original Diffusion Policy paper, the U-Net based diffusion policy, to faithfully evaluate our baselines. Notably, the original U-Net based diffusion policy entails over 80M parameters, while the specialist employed in RoboDual has merely 17M. We designed this lightweight specialist mainly to enable higher frequency control in the dual-system framework.
>
> During the rebuttal period, we did additional experiments with our specialist-only model as a baseline for the "Put Block into Bowl" task, and the results are shown below:
>
> |Methods|Success Rate|
> |-|-|
> |Specialist-only (DiT)|40%|
> |Diffusion Policy|53.3%|
> |RoboDual (Ours)|93.3%|
>
> We would like to highlight that our performance improvements come primarily from the framework of dual-system synergy, rather than from improvements in the generalist and specialist individually (though RoboDual can be applied to the more powerful generalist and specialist to achieve further improvements).
>
> > *${\color{BrickRed}Q7:}$*  Is the OpenVLA in your experiments the same model used as the generalist in RoboDual (generalist only)?
>
> The architecture is the same. However, the generalist in RoboDual, as specified in Section 4.1, is only trained on the mixture of our real-world robot data in a multi-task learning setting. While OpenVLA, serving as a baseline in our experiments, is first trained on all task data and then finetuned on each task to optimize its performance (as we found that performing only multi-task learning on OpenVLA leads to even lower performance).

---

> ### Author Response · Authors · 2024-12-02
> **Looking forward to your prompt response**
>
> We sincerely hope that we have addressed all of your concerns satisfactorily. **As the rebuttal phase is about to conclude**, we would greatly appreciate your prompt response. Please feel free to share any further comments or concerns you may have.

---

### Author Response · Authors · 2024-11-21
**General Author Response for Rebuttal**

Dear Area Chairs and Reviewers,

We thank all the Reviewers for their detailed and helpful comments on our work. We appreciate the Reviewers for acknowledging our strengths and contributions, such as an innovative (woVN, Sftm), compeling (AWwV), and insightful (Sftm) pipeline to effectively leverage generalist's generalization and specialist's efficiency (woVN, bxCB, AWwV), efficient training and inference for practical applications (bxCB, AWwV), extensive experiments (Sftm) and significant improvements (woVN, bxCB, AWwV, Sftm), and well-written (woVN) and highly informative figures (Sftm).

During the rebuttal phase, we have made diligent efforts to address the concerns raised by the Reviewers, add more real-world tests on the generalizaiton ability, provide discussions on technical details, and improve color maps in figures and clarity of various expressions. We have carefully made corresponding modifications (highlighted in blue) in the updated manuscript. Our responses to specific concerns are detailed below. We thank you all for the opportunity to improve our work with your constructive feedback.

Best regards,
The Authors

---

> ### Comment · Area_Chair_EQza · 2024-11-25
> **Reviewer Response**
>
> Dear Reviewers,
>
> The rebuttal discussion period is coming to an end and the authors
> have spent a large amount of type responding to each concern. Can
> everyone look at the reviews and let the authors know if there are
> any remaining concerns?
>
> Best,
>
> AC

---

### Meta-Review · Area_Chair_EQza · 2024-12-20

**Metareview:**

This paper investigates how to combine the generalization capability of models like OpenVLA with the accuracy and task-specific precision of methods such as ACT or Diffusion Policy. To address this, the authors propose a novel framework, RoboDual, which uses the intermediate tokens in OpenVLA to condition a modified Diffusion Policy. Through simulation and real-world experiments, the proposed method demonstrates substantial performance improvements over both generalist and specialist baselines.

Overall, the strength of the paper is that it proposes an interesting and pertinent research question. However, a weakness of the paper is that while it trys to bridge the generalization ability of model like VLA with that of task-specific diffusion policy, the actual results end up being not very dexterous, not really illustrating the effectiveness of this approach.

**Additional Comments On Reviewer Discussion:**

There were no reviewers who championed the paper, with the reviewer who leaned to rejecting the paper give valid issues that the authors ultimately did not resolve.

---

### Decision · Program_Chairs · 2025-01-22

Reject